# Information transmission in a cell monolayer: A numerical study

**Paweł Nałęcz-Jawecki**[1], **Przemysław Szyc**[2], **Frederic Grabowski**[1],
**Marek Kochańczyk**[1], **Tomasz Lipniacki**[1*]

**1** Institute of Fundamental Technological Research, Polish Academy of Sciences, Warsaw, Poland,
**2** Faculty of Physics, University of Warsaw, Warsaw, Poland

\* tlipnia@ippt.pan.pl

pcbi.1012846

STATES OF AMERICA

**Peer Review History:** PLOS recognizes the
benefits of transparency in the peer review
process; therefore, we enable the publication
of all of the content of peer review and
author responses alongside final, published
articles. The editorial history of this article is
available here: https://doi.org/10.1371/journal.
pcbi.1012846

## Abstract

Motivated by the spatiotemporal waves of MAPK/ERK activity, crucial for long-range communication in regenerating tissues, we investigated stochastic homoclinic fronts propagating through channels formed by directly interacting cells. We evaluated the efficiency of long-range communication in these channels by examining the rate of information transmission. Our study identified the stochastic phenomena that reduce this rate: front propagation failure, new front spawning, and variability in the front velocity. We found that a trade-off between the frequencies of propagation failures and new front spawning determines the optimal channel width (which geometrically determines the front length). The optimal frequency of initiating new waves is determined by a trade-off between the input information rate (higher with more frequent initiation) and the fidelity of information transmission (lower with more frequent initiation). Our analysis provides insight into the relative timescales of intra- and intercellular processes necessary for successful wave propagation.

## Author summary

In biological tissues, traveling waves of cellular activity are observed in the process of wound healing when they coordinate cell replication and collective migration. These waves can carry information over long distances. However, random effects on the single-cell level can affect wave propagation and disrupt information flow. In this paper, using a numerical model we classified these stochastic events and quantified the maximum range and frequency of such waves and their capacity to carry information. We discovered that most effective transmission occurs in relatively narrow channels (formed by directly interacting cells), and that the refractory time, in which a cell is resistant to activation by neighboring cells, must be long with respect to the time needed for cell activation. The optimal time intervals between the initiated waves are of order of few refractory times (depending on channel length).

## Introduction

Cells in living organisms communicate through a variety of mechanisms, including chemical and mechanical signals. Long-range communication within a tissue may result from local

**Data availability statement:** The simulator code is available at https://github.com/kochanczyk/qeirq. The code that can be used to reproduce all reported results is available at https://github.com/pszyc/visavis-seir.

**Funding:** This work was supported by the National Science Centre Poland (https://ncn.gov.pl) grant 2019/35/B/NZ2/03898 to T.L. The funders had no role in study design, data collection and analysis, decision to publish, or preparation of the manuscript.

communication between neighboring cells. This is the case for spatiotemporal MAPK/ERK activity waves, originating from the wound edge [1,2] or from leader cells [3] and involving a mechanochemical feedback loop that coordinates collective migration of epithelial cells [3,4]. Excitable ERK activity waves have been shown to control the rate of scale regeneration in zebrafish [5]. These studies highlight the capability of waves to propagate recurrently across successive cell layers despite the inherent discreteness and heterogeneity of the communication medium.

From a dynamical systems perspective, a traveling front may be either an interface between regions in space that are in different equilibria (heteroclinic traveling waves in bistable systems) or an excitation that locally departs and then returns to a unique equilibrium (homoclinic traveling waves in monostable excitable systems) [6]. The heteroclinic traveling waves are formed robustly at the interface between two different regions and as such are resilient to random perturbations [7]. However, the passage of a heteroclinic wave irreversibly changes the state of the reactor that consequently cannot be re-used to support propagation of a subsequent front. In contrast, a homoclinic traveling wave is an out-of-equilibrium "stripe" flanked on both sides with the reactor in the equilibrium state. Although homoclinic traveling waves may be sent recurrently and in this way convey complex messages to spatially distant locations, they are fragile in the presence of stochastic fluctuations. Experimentally, the spatiotemporal ERK signal propagation has been observed to be distorted by random bursts of ERK activity [2,8], which occasionally give rise to spontaneous waves [9].

Here, we sought to quantitatively assess the capacity of a discrete, excitable medium to transmit information encoded within a train of activity waves. Specifically, we investigated stochastic homoclinic fronts propagating in narrow channels formed by directly interacting cells, and determined the efficiency of long-range communication through these channels in terms of the rate of information transmission, also referred to as bitrate. This metric quantifies the amount of information that can be transmitted through a communication channel in a unit of time. We identified several types of stochastic phenomena that reduce the fidelity and thus the rate of information transmission, among which the front propagation failure, new front spawning, and variability in the front velocity were the most impactful. We demonstrated that a tradeoff between the frequencies of front propagation failure and new front spawning determines the optimal channel width, enabling the fronts to reach the greatest distance and maximizing the rate of information transmission. We investigated the system's ability to relay periodic sequences of fronts as well as transmit binary-encoded information. Binary-encoded information is encoded by specifying a predetermined list of equally spaced time slots and deciding whether a front is initiated or not in each of these time slots. We determined the time interval between the time slots that maximizes the information transmission rate. This optimal time interval or, equivalently, the optimal frequency of time slots results from a tradeoff between the input information rate (higher for on average more frequently initiated fronts) and fidelity of information transmission (lower for on average more frequently initiated fronts). Finally, our exploration of the model parameter space revealed that efficient long-distance information transmission is achievable only if the refractory time is several times longer than the neighbor-to-neighbor activation time.

## Results

### Model definition

In our continuous-time discrete-space model, each cell in the monolayer assumes one of four distinct states: quiescent (Q), excited (E), inducing (I), or refractory (R), and inter-state transitions follow a predetermined cyclic sequence: Q → E → I → R → Q (Fig 1A). When activated by an

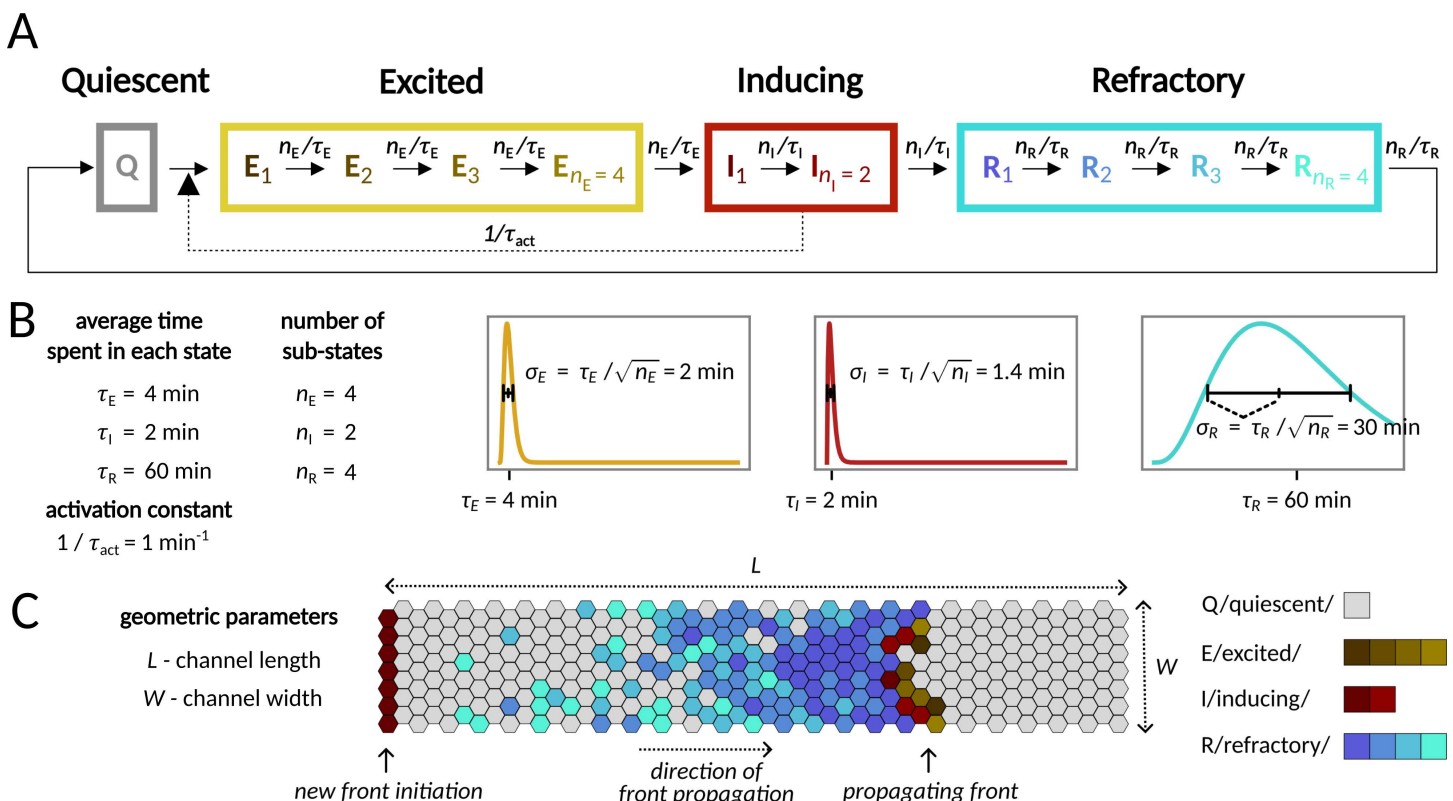

**Fig 1. Model. A** Scheme of the QEIRQ model with multiple sub-states. A cell in one of the Inducing states ($I_1$ or $I_2$) can activate an adjacent cell in the Quiescent state (Q) turning its state to the first Excited state ($E_1$). Transitions between sub-states are spontaneous first order reactions. **B** Parameters of the model and their nominal values. Total time spent in each of the functional state (E, I, R) has Erlang distribution with scale $n_E$, $n_I$, $n_R$ and rate $n_E/\tau_E$, $n_I/\tau_I$, $n_R/\tau_R$, respectively, plotted in the graphs. The distributions have means $\tau_E$, $\tau_I$, $\tau_R$ and standard deviations $\tau_E/\sqrt{n_E}$, $\tau_I/\sqrt{n_I}$, $\tau_R/\sqrt{n_R}$, respectively, as indicated in the figures. **C** Geometry of the reactor used for simulations. Periodic boundary conditions are applied along the longer edge (unless explicitly stated otherwise). A new front is initiated by setting the leftmost layer of cells to the first Inducing state ($I_1$).

I neighbor, a Q cell assumes the E state. Subsequent progression from the E to the I state enables the cell to induce activity in its Q neighbors. After the I cell assumes the R state, it loses the ability to activate its Q neighbors and becomes insensitive to activation by neighboring I cells. Finally, reverting to the Q state restores the cell's responsiveness to activation by an I neighbor.

Sequential transition through Q, E, I, and R states and appropriate time scales of the residence in the E, I, and R states are crucial to enable wave propagation. In the context of the MAPK/ERK pathway, the first time scale is associated with signal reception and the signal transduction cascade (through the EGFR/SOS/RAS complex, RAF, and MEK), culminating in ERK phosphorylation; the second time scale is determined by the time required by phospho-ERK to trigger cell contraction that may result in activation of EGFR in neighboring cells [3]; and the third, longest, time scale is related to the refractory period of the signaling cascade, which, due to inhibitory multisite phosphorylation of SOS (by phospho-ERK), is at least partially insensitive to incoming signals.

In the model, the multi-step signal transduction within the MAPK/ERK pathway is reflected by the assumption that the E, I, and R states comprise multiple sub-states ($n_E$, $n_I$, $n_R$, respectively). We assume that each transition between the sub-states (e.g., $E_1 \to E_2$) is Markovian, and thus the transition times are exponentially distributed (with the rate parameters $n_E/\tau_E$, $n_I/\tau_I$, or $n_R/\tau_R$; nominal values of the kinetic parameters are provided in Fig 1B). Consequently,

the total residence times in the E, I, and R states follow Erlang distributions: Erlang($n_E$, $n_E/\tau_E$), Erlang($n_I$, $n_I/\tau_I$), and Erlang($n_R$, $n_R/\tau_R$) with means $\tau_E$, $\tau_I$, and $\tau_R$, respectively. Note that by increasing $n_E$, $n_I$, or $n_R$ we reduce the stochasticity of the system, because the variance of an Erlang($n$, $n/\tau$) distribution is inversely proportional to $n$.

In the simulated fully-confluent monolayer, cells are immobile agents arranged on a 2-D triangular lattice of length $L$ and width $W$ (Fig 1C). For the nominal parameter values (Fig 1B), the rates of transitions Q → $E_1$ ($1/\tau_{act}$), $I_1$ → $I_2$, and $I_2$ → R (both $n_I/\tau_I$) are all equal. Thus, the probability that an I cell will activate a given neighboring Q cell before transitioning to R is $0.5 + 0.5 \times 0.5 = 0.75$. This value is greater than both the site and the bond percolation thresholds on the triangular lattice, equal 0.5 and ~0.347, respectively [10], permitting front propagation [11].

In the model, we formally assume that all cells are identical and all variability follows from stochastic transitions between cell states. In recent years, it has been demonstrated that population heterogeneity is mainly responsible for observed variability in signal transduction [12–14]. However, in our case (as in many other cases) the two types of noise (intrinsic and extrinsic) have similar consequences. Stochasticity in front propagation is a consequence of variability in state-to-state transition times; this variability may have both intrinsic and extrinsic origins. The key difference is that under intrinsic noise, the front is expected to be perturbed in random locations, while under extrinsic noise, some specific locations in the monolayer will more likely be the source of front perturbation.

## Propagation of a single front

### Front speed

A propagating front consists of active cells (i.e., cells in either state E or state I) located in its head, followed by a thick block of R cells that prevent backward front propagation. We initiated fronts by setting the states of the cells in the first layer to $I_1$ and observed the propagation of activity toward the other end of the reactor (Fig 1C).

In the model, the front is deemed to move one step forward once the next cell layer is activated and progresses through all E sub-states to become I. On average, a forward step, in which the front advances by just one cell layer, takes the time $\tau_{act}/\langle n_{neigh\_I}\rangle + \tau_E$, where $\langle n_{neigh\_I}\rangle$ is the average number of I cells in contact with a single Q cell. In a deterministic model, where the time spent by cells in each state is fixed, the front forms a straight line, and each Q cell at the front head has exactly 2 inducing neighbors. Thus, the inverse propagation speed is $v^{-1}_{deterministic} = \tau_{act}/2 + \tau_E$, which yields 4.5 min/cell layer for the nominal parameter values.

In the stochastic model, however, the front edge becomes rough and the cells to be activated have, on average, more than two I neighbors, which increases the propagation speed. We observed that for the nominal parameter values, the average inverse front velocity $\langle v^{-1}\rangle$ changes with the channel width $W$ from $v^{-1}_{deterministic}$ for $W = 1$ to the asymptotic value $v^{-1}_{asymptotic} = 3.5$ min/layer for $W \gtrsim 10$ (Fig A panel a in S1 Appendix).

### Transit time and its variance

The expected time in which the front travels the whole channel length $L$ is $\langle \tau_{transit}\rangle = L\langle v^{-1}\rangle$. For sufficiently long channels, the distribution of $\tau_{transit}$ is nearly Gaussian with variance $\sigma^2_{transit} = L \times \sigma^2_0(W)$. The value of $\sigma^2_0$ decreases with the channel width, and for W > 2 can be well approximated with the formula $\sigma^2_0 = a/W + b$ with $a \approx 6.8$ min$^2$, $b \approx 0.4$ min$^2$/layer (Fig A panel b in S1 Appendix). The variance $\sigma^2_{transit}$ critically affects the fidelity of information transmission by determining the precision with which the moment of a front initiation can be inferred from the time at which it reached the end of the channel.

## Propagation failure and front spawning

In a stochastic model, a traveling front is subjected to random events disrupting its propagation. In narrow and moderately wide channels, we observed two types of disruptive events: *propagation failure* and *new front spawning* (Fig 2A).

Propagation fails if all E and I cells progress to the R state before exciting any neighboring Q cells (S1 Video). New fronts are spawned when a cell remains in the I state long enough for one of its neighboring cells to recover from R to Q. Such a neighboring cell may get activated and become a source of a new front or fronts (S2 Video). The new front(s) can propagate backward or forward. When a backward-propagating front encounters a forward-propagating front, they collide and usually annihilate (Fig 2B and S3 Video) or, rarely, give rise to another front. For broad reactors (>20 cells wide), fronts may propagate in directions not necessarily parallel to the channel longer axis (Fig B in S1 Appendix and S4 Video), which leads to a chaotic front pattern characteristic for the Greenberg–Hastings model (first defined in Ref. [15] and later recast as a stochastic model and studied, e.g., in Ref. [16]). This 2-dimensional effect is not observed in narrow channels, in which the front tail (the block of cells in the R state at the front's rear side) is longer than the channel width.

We found that the propensity (probability per one cell layer) of a propagation failure event decreases with the channel width $W$ as

$$\lambda_{\text{fail}} = \exp\left(a_{\text{fail}} \times \left(W - W_{\text{fail}}\right)\right), \tag{1}$$

where $a_{\text{fail}} < 0$ and $W_{\text{fail}}$ are coefficients that depend on model parameters (Fig 2C). The exponential dependence on $W$ results from the fact that the number of cells in either the E or the I state is proportional to the channel width, and for a front to disappear *all* of them have to simultaneously progress to R without exciting new cells. In contrast, the propensity of new front spawning event increases linearly with channel width as

$$\lambda_{\text{spawn}} = a_{\text{spawn}} \times \left(W - W_{\text{spawn}}\right), \tag{2}$$

with $a_{\text{spawn}}$ and $W_{\text{spawn}}$ dependent on the model parameters. This is because spawning may be triggered by *any* cell across the channel width (Fig 2C).

We noticed that in narrow channels ($W \leq 6$), a single backward front is usually spawned (Fig 2D). In contrast, in broader channels ($6 < W \leq 10$), multiple fronts are often created in a single spawning event. The fronts are typically spawned alternately backward and forward, implying a correlation between the number of fronts generated in either direction. For even broader channels ($W > 10$), long-lasting spawning sites can generate new fronts for prolonged periods, blocking any information transmission. This becomes a dominant disruptive event in very broad channels of $W > 20$ (Fig 2E).

## Optimal channel width

In the case of a narrow channel transmitting a series of fronts, the impact of the two types of disruptive events is similar. Propagation failure eliminates the front, whereas a backward-spawned front collides with and annihilates the subsequent (forward-propagating) front. In both scenarios, the total number of fronts received at the other side of the channel is reduced by 1. In narrow channels, a spawning event usually generates a single backward front. Thus, front elimination in a given time span can result either from its propagation failure or from a backward front spawned by the preceding front. We assume that $\lambda_{\text{fail}}$ is small enough so that the probability that the backward front disappears before collision is negligible. Therefore, in narrow channel the total front elimination propensity is roughly

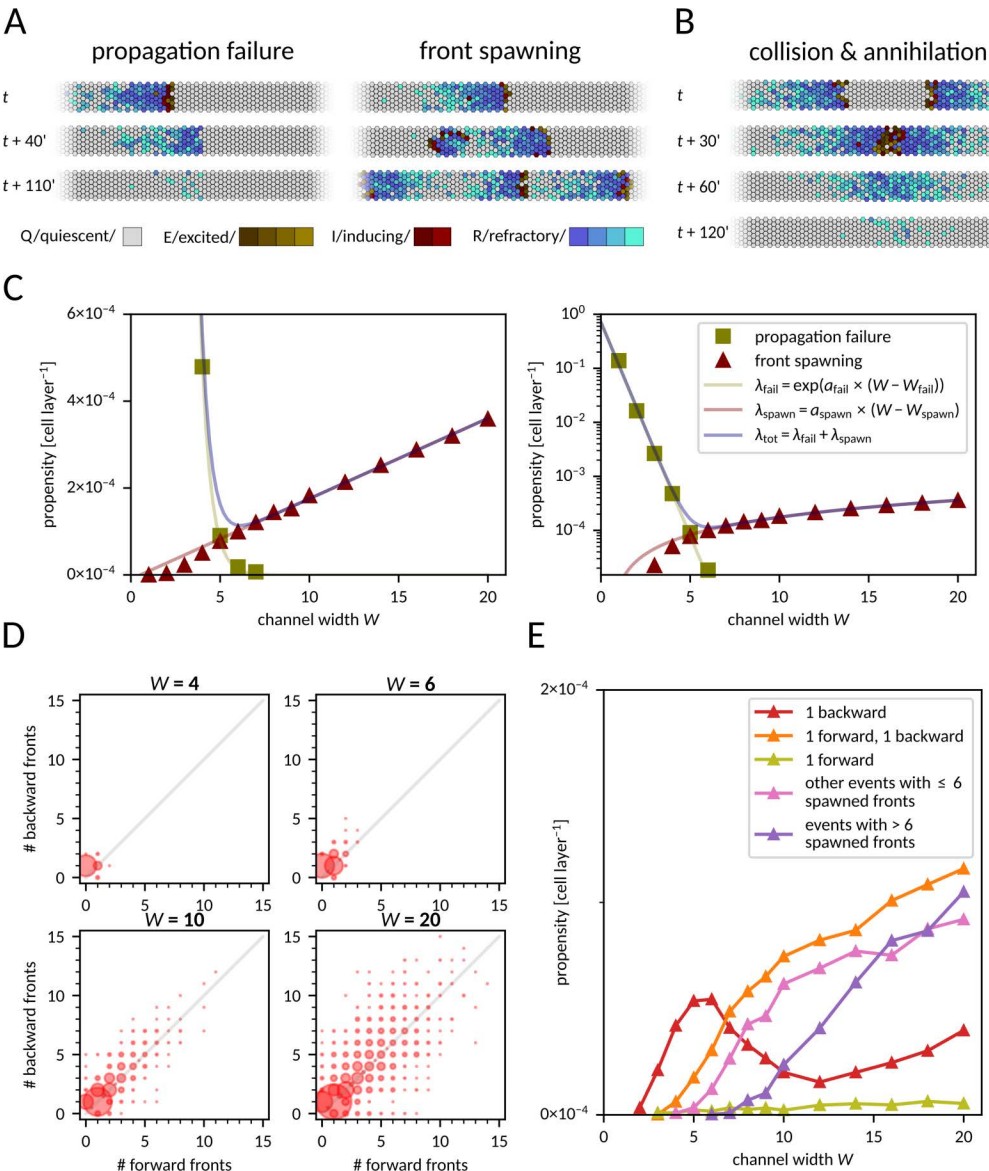

**Fig 2. Disruptive events associated with front propagation. A** Snapshots from simulations with the two kinds of disruptive events. Left: front dies out as all Excited or Inducing cells turn Refractory before exciting adjacent cells. Right: a single Inducing cell persists long enough for one of its neighboring cells to revert to the Quiescent state. This neighboring cell can then be reactivated, becoming a source of new fronts (in this case, one forward and one backward). See also S1 and S2 Video for full time-courses. **B** Two fronts propagating in opposite directions collide and annihilate. See also S3 Video. **C** Propensity (probability per cell layer) of disruptive events as a function of channel width, in linear (left) and log (right) scale. Linear functions fitted to $\ln(\lambda_{fail})$ and $\lambda_{spawn}$ have coefficients $a_{fail} = -1.82$, $W_{fail} = -0.19$, $a_{spawn} = 1.85 \times \times 10^{-5}$, $W_{spawn} = 0.53$. **D** Distribution of the number of forward and backward fronts spawned from a single localized spawning site for four different channel widths. Each disk area is proportional to the probability that a corresponding number of backward and forward fronts was spawned in a single event; the total area of all disks is proportional to the front spawning propensity, which is different for each channel width. **E** Propensity of spawning one or more fronts in a single localized disruptive event as a function of channel width. Data for panels C–E was gathered from 30,000 simulations for each channel width, channel length was fixed at $L = 300$.

$$\lambda_{\text{tot}} = \lambda_{\text{fail}} + \lambda_{\text{spawn}}. \tag{3}$$

As $\lambda_{\text{fail}}$ decreases exponentially and $\lambda_{\text{spawn}}$ increases linearly with the channel width, there is an optimal width at which $\lambda_{\text{tot}}$ is minimized. This optimal width maximizes the probability that a given front in a front series passes uninterruptedly through the channel. For the nominal parameter values and periodic boundary conditions along the longer channel edges, the optimal channel width is $W_{\text{opt}} = 6$ (Fig 2C), and for this width the average range of uninterrupted front propagation is about 8000 cell layers. As we will see later, the optimum is robust to moderate changes in the values of model parameters. We thus chose the value $W_{\text{opt}} = 6$ as the default channel width for most of the paper. Since uninterrupted front propagation is critical for information transmission, we expected that the information transmission rate would be highest when $W$ is close to $W_{\text{opt}}$.

For non-periodic (inert) boundary conditions, for which interacting cells are bordered by biochemically inert cells, both $\lambda_{\text{fail}}$ and $\lambda_{\text{spawn}}$ are substantially larger because of disruptions initiated at the edges of the reactor. This modifies $\lambda_{\text{spawn}}$ by a constant additive term. Consequently, the minimum value of $\lambda_{\text{tot}}$ is about four-fold higher and attained at $W = 7$ (Fig C in S1 Appendix). As it will be showed later, higher propensities of disruptive events imply lower information transmission rate.

## Interaction between fronts

### Refractory time

The tail of a front consists of a block of refractory cells, which hinder propagation of the next front. A cell requires, on average, $T_{\text{cycle}} = \tau_{\text{act}}/\langle n_{\text{neigh\_I}} \rangle + \tau_{\text{E}} + \tau_{\text{I}} + \tau_{\text{R}} = 66.5$ min to make a full cycle from Q through E, I, and R back to Q (assuming default parameter values and $\langle n_{\text{neigh\_I}} \rangle = 2$). In the deterministic model, during this period, the channel is fully blocked; only fronts initiated at a rate smaller than $1/T_{\text{cycle}}$ are transmitted. In the stochastic model, $T_{\text{cycle}}$ defines the time scale of the inter-front interval, below which the propensity of disruptive events is significantly elevated, as one can see in the kymographs in Fig 3A.

To investigate how often fronts can be initiated and reliably transmitted, we performed simulations in which pairs of fronts were initiated at various inter-front intervals (Fig 3B). As expected, for intervals shorter than $T_{\text{cycle}}$, propagation of the second front typically fails immediately as the cells at the beginning of the channel have not recovered to Q yet (immediate failure probability is 50% for the interval of 61.5 min). The proximity of the previous front also markedly increases the probability of front spawning. This is likely due to some R cells remaining after the passage of the first front, which recover and become Q only after the passage of the second front. These cells may get activated within the block of R cells in the second front's tail, seeding a new front at the second front's rear side.

The described effects were found to be most significant for inter-front intervals in the range of 50–130 min, i.e., around the value of $T_{\text{cycle}} + \sigma_{\text{cycle}} \approx 96.6$ min, where $\sigma^2_{\text{cycle}} = (\tau_{\text{act}}/\langle n_{\text{neigh\_I}} \rangle)^2 + \tau_{\text{E}}^2/n_{\text{E}} + \tau_{\text{I}}^2/n_{\text{I}} + \tau_{\text{R}}^2/n_{\text{R}} \approx (30.1 \text{ min})^2$ is the variance of the time of the full cycle $Q \rightarrow E \rightarrow I \rightarrow R \rightarrow Q$. The time $T_{\text{cycle}} + \sigma_{\text{cycle}} =: T_{\text{R}}$ can be considered the effective refractory time. For inter-front intervals longer than 130 min, the propensities of disruptive events approach the single-front values (shown in Fig 2C). For intervals shorter than 50 min, the propensities of disruptive events remain high, but the overall probability of their occurrence becomes low, due to the high chance of immediate failure, which we count separately.

### Propagation of periodically initiated fronts

The refractory time forces subsequent fronts to not travel too closely, and consequently the average interval between fronts reaching the end of the channel has a minimum (with respect

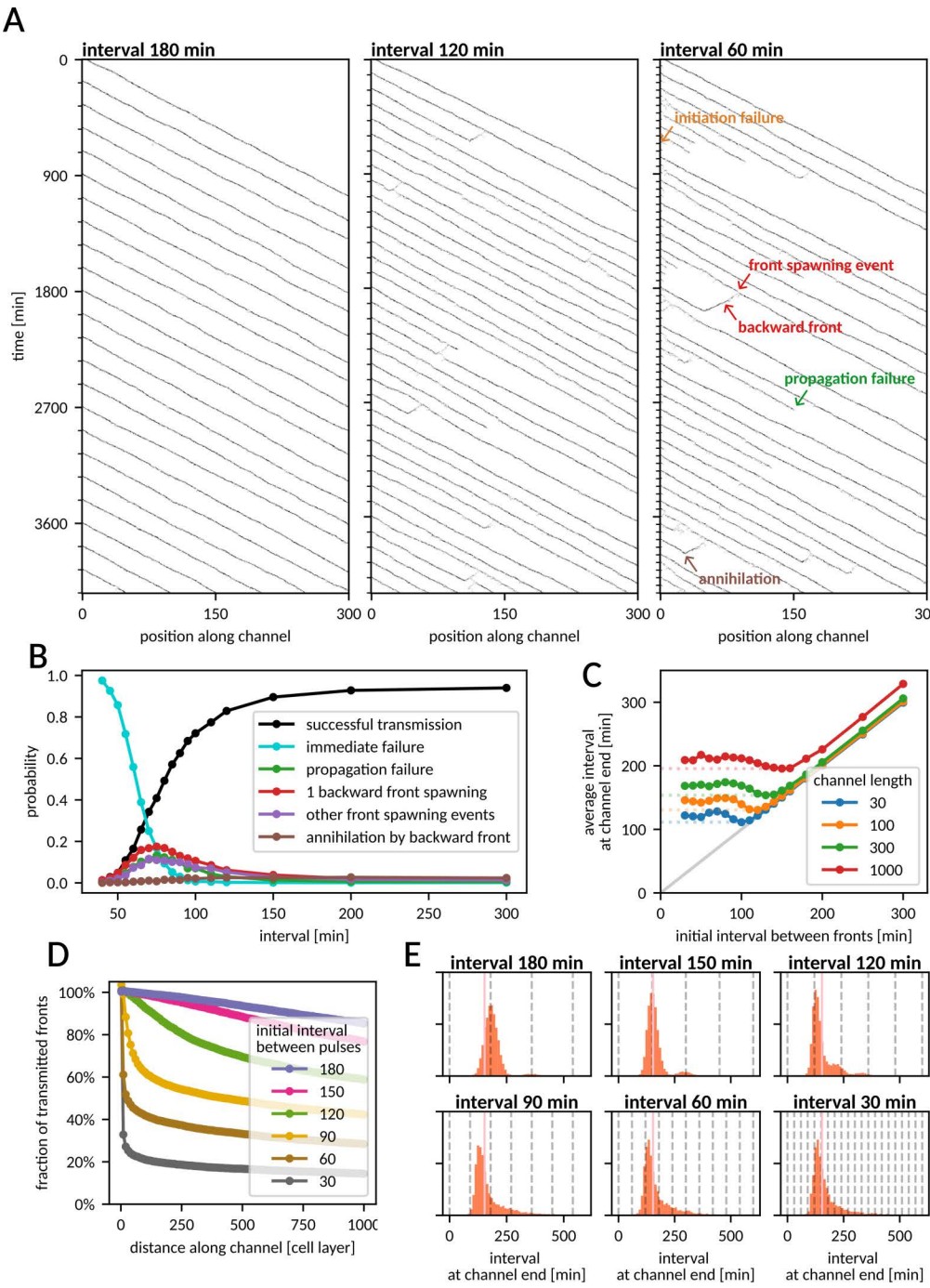

**Fig 3. Occurrence of disruptive events as a function of the interval between fronts. A** Kymographs of the number of active cells (E or I) as a function of time and distance along the channel or fronts initiated periodically at different intervals. Channel dimensions: $W = 6$, $L = 300$. **B** Probability of disruptive events for the latter of two fronts, as a function of the (initial) inter-front interval. Channel dimensions: $W = 6$, $L = 300$. **C** Average interval between fronts reaching the end of the channel as a function of the initial inter-front interval for different channel lengths. For each channel length, $T_{trans-min}$ is marked with dotted lines. Channel width $W = 6$, length as indicated. **D** Percentage of fronts that reached a certain distance along the channel for different initiation frequencies. Channel dimensions: $W = 6$, $L = 1000$. **E** Distribution of the interval between fronts that reached the end of the channel for different initial intervals. Dashed vertical lines denote time points at which fronts were initiated. Pink vertical line shows $T_{trans-min}$. Channel dimensions: $W = 6$, $L = 300$. In all panels, channel width $W = 6$. Data for panel B from 3000 simulations for each data point; data for panels C–E from 30 simulations of 500 fronts for each interval.

to the frequency of front initiation), which we denote $T_{\text{trans-min}}(L)$. In a short channel of length $L = 30$, the effective refractory time $T_{\text{R}}$ is a good approximation of $T_{\text{trans-min}}$ (Fig 3C and Fig D in S1 Appendix). In a long channel, $L = 1000$, we observe that $T_{\text{trans-min}}$ is about 200 min and is achieved when fronts are initiated every 160 min, while a more frequent front initiation results in a slightly longer average front arrival interval. In long channels, a considerable percentage of fronts sent at periods shorter than $T_{\text{R}}$ is eliminated shortly after initiation (Fig 3D). For fronts sent every 150 or 180 min, the propensity of front elimination is initially low, but increases slightly with the distance from the initiation site (Fig 3D). This is because for such intervals disruptive events are relatively rare, but due to fluctuating velocities some fronts draw closer together, and thus the likelihood of disruptive events increases. Consequently, in a channel of length $L = 300$, fronts initiated every 150 or 180 min arrive with time spans distributed around the initial period (Fig 3E, top row), whereas fronts sent with periods $< T_{\text{R}}$ arrive with time spans distributed more broadly around $T_{\text{trans-min}}$ regardless of the initial pulse frequency (Fig 3E, bottom row). Overall, we found that in the limit of short channel lengths, the empirically estimated refractory time $T_{\text{R}}$ sets an upper limit on the transmittable front frequency, while in longer channels, the maximum transmission frequency is lower, and lower initiation frequencies enable propagation of more coherent front trains.

## Information transmission rate

### Numerical results

To estimate the bitrate, we used a simple binary protocol in which a sequence of equiprobable (probability of a '1' $q = \frac{1}{2}$) binary symbols $S_i \in \{0, 1\}$ is directly translated into a sequence of fronts (1 → front initiated, 0 → no front initiated) and sent through the channel at regular time slots, with inter-slot interval $T_{\text{slot}}$. We performed simulations and registered the time points at which the fronts reached the end of the reactor. For each time slot $t_{\text{slot}}$, we computed the expected arrival time $t_{\text{expected}} = t_{\text{slot}} + \langle \tau_{\text{transit}} \rangle = t_{\text{slot}} + L \langle v^{-1} \rangle$ and selected the front that arrived closest to $t_{\text{expected}}$. This could be a successfully transmitted front initiated in the considered slot, a front sent in another slot, or a front spawned in a disruptive event. We recorded the difference $\Delta t = t_{\text{arrival}} - t_{\text{expected}}$ between the closest front's arrival time $t_{\text{arrival}}$ and $t_{\text{expected}}$. Using all slots in the sequence, we estimated the mutual information per slot ($\text{MI}_{\text{slot}}$) as mutual information between $\Delta t$ and the corresponding binary symbol $S$ (see Methods for details). In Fig E in S1 Appendix we show histograms of $\Delta t$ for S = 1 and S = 0. For $T_{\text{slot}} = 150$ min these histograms are well separated, while for $T_{\text{slot}} = 90$ min there is a significant overlap. This implies a higher $\text{MI}_{\text{slot}}$ for $T_{\text{slot}} = 150$ min. Finally, we calculated the information transmission rate $r = \text{MI}_{\text{slot}} / T_{\text{slot}}$. The results for various channel lengths and inter-slot intervals are presented in Fig 4A.

As one can observe, information transmission is highest for moderate values of $T_{\text{slot}}$. For long $T_{\text{slot}}$, the fraction of successfully transmitted information (equal to $\text{MI}_{\text{slot}}$ measured in bits, as each binary symbol carries one bit of information) is determined by the distant-front dynamics and is thus roughly independent of $T_{\text{slot}}$ (Fig 4B). For large values of $T_{\text{slot}}$ we may define two regimes:

(1) The free-front regime, in which the interval between slots is at least twice longer than the transit time. In this regime, fronts spawned backward cannot collide with subsequent fronts, because they reach the beginning of the channel and disappear before the subsequent front is initiated. Consequently, information transmission is determined by the single-front propagation failure propensity. This regime is characteristic of short channels or very long inter-slot intervals.

(2) The distant-front regime, in which fronts are frequent enough to be annihilated by backward fronts spawned by their predecessors, yet still maintain sufficient distance to avoid

direct interaction (for the default parameters and $L = 1000$ this means $T_{slot} > 200$ min, see Fig 3C). In this regime, the propensity of disruptive events is still equal to that observed for single fronts, but information transmission is limited by the total disruptive event propensity, as both propagation failure and backward front spawning lead to extinction of one forward front.

For the considered set of parameters, information transmission efficiency ($MI_{slot}$) is substantially higher in the free-front than in the distant-front regime, because the annihilation of forward-propagating fronts by spawned backward-propagating fronts (Fig 2D) is the main limiting factor in the distant-front regime. In both regimes, $MI_{slot}$ is nearly independent of $T_{slot}$, and thus the bitrate (equal to $MI_{slot}/T_{slot}$) decreases as $1/T_{slot}$. On the other hand, for short inter-slot intervals the bitrate is limited by strong interactions between fronts (increasing propensity of disruptive events) and transit time dispersion comparable to $T_{slot}$. Thus, for each channel length, there is an optimum inter-slot interval $T_{opt}$ for which the bitrate is the highest.

For the nominal parameter values and channel length $L = 30$, the optimum is located at $T_{opt} \approx T_R \approx 96.6$ min, and the maximum bitrate is ~0.5 bit/h. Unsurprisingly, the bitrate decreases with an increasing channel length $L$ regardless of $T_{slot}$ due to the accumulation of disruptive effects (Fig 4A and 4C). The optimal interval $T_{opt}$ increases with $L$, and in the investigated range of channel lengths, the increment is roughly proportional to $\sqrt{L}$ (Fig 4D), which can be attributed to accumulation of transit time variance.

The maximum information rate depends on the channel width (Fig 4E). As expected, the highest bitrate is observed for $W = 6$, for which the total disruptive event propensity is lowest (as shown Fig 2C). In short channels ($L = 30$), bitrate is a slowly decreasing function of $W$. On the contrary, in longer channels ($L \geq 100$), bitrate decreases faster and drops nearly to zero when long-lasting spawning blocks any regular front propagation. This implies that in the tissue, information can reach relatively short distances from a wound, unless the fronts are confined to narrow structures, such as capillary vessels. The optimal inter-slot interval grows with $W$ (Fig 4F), indicating that in broader channels, in which spawning events are more likely, the reliability of information transmission gained by increasing the distance between fronts (which reduces spawning propensity, Fig 3B) is worth the cost of the reduced front frequency.

## Semi-analytical predictions based on phenomenological analysis

There are two major phenomena that set bounds on the information transmission rate: disruptive events and transit time stochasticity. If the intervals between slots are long, the major limiting factor is the possibility of front extinction, due to either propagation failure or a collision with a spawned backward-propagating front. In this case, the amount of transmitted information per slot can be obtained from the confusion matrix with the formula $MI_{slot} = 1 - 1/2 \times [(p + 1) \log_2 (p + 1) - p \log_2 p]$, where $p$ is the probability of front extinction (see Methods for details). To determine the value of $p$, we used the propensities $\lambda_{fail}$ and $\lambda_{spawn}$ of disruptive events for single fronts (Fig 2C) and estimated the expected number of backward fronts spawned in a single event based on results shown in Fig 2D. We were then able to obtain a satisfactory prediction of $MI_{slot}$ in the distant-front limit ($T_{slot} \to \infty$) as shown in Fig 4B. By replacing the distant-front event probabilities with estimates for finite interval between fronts from Fig 3B we obtained satisfactory predictions of the bitrate for $T_{slot} > 120$ min for $L = 100$ and $L = 300$ and for $T_{slot} > 200$ min for $L = 1000$ (Fig 4G–4I, dotted line).

The prediction taking into account only the disruptive events is satisfactory for long intervals and/or short channels. When $T_{slot}$ is comparable to or shorter than the standard deviation of the transit time, $\sigma_{transit}$, fronts reaching the end of the channel may be assigned to a wrong slot. The value of $\sigma_{transit}$ scales proportionally to $\sqrt{L}$, which makes the misassignment more

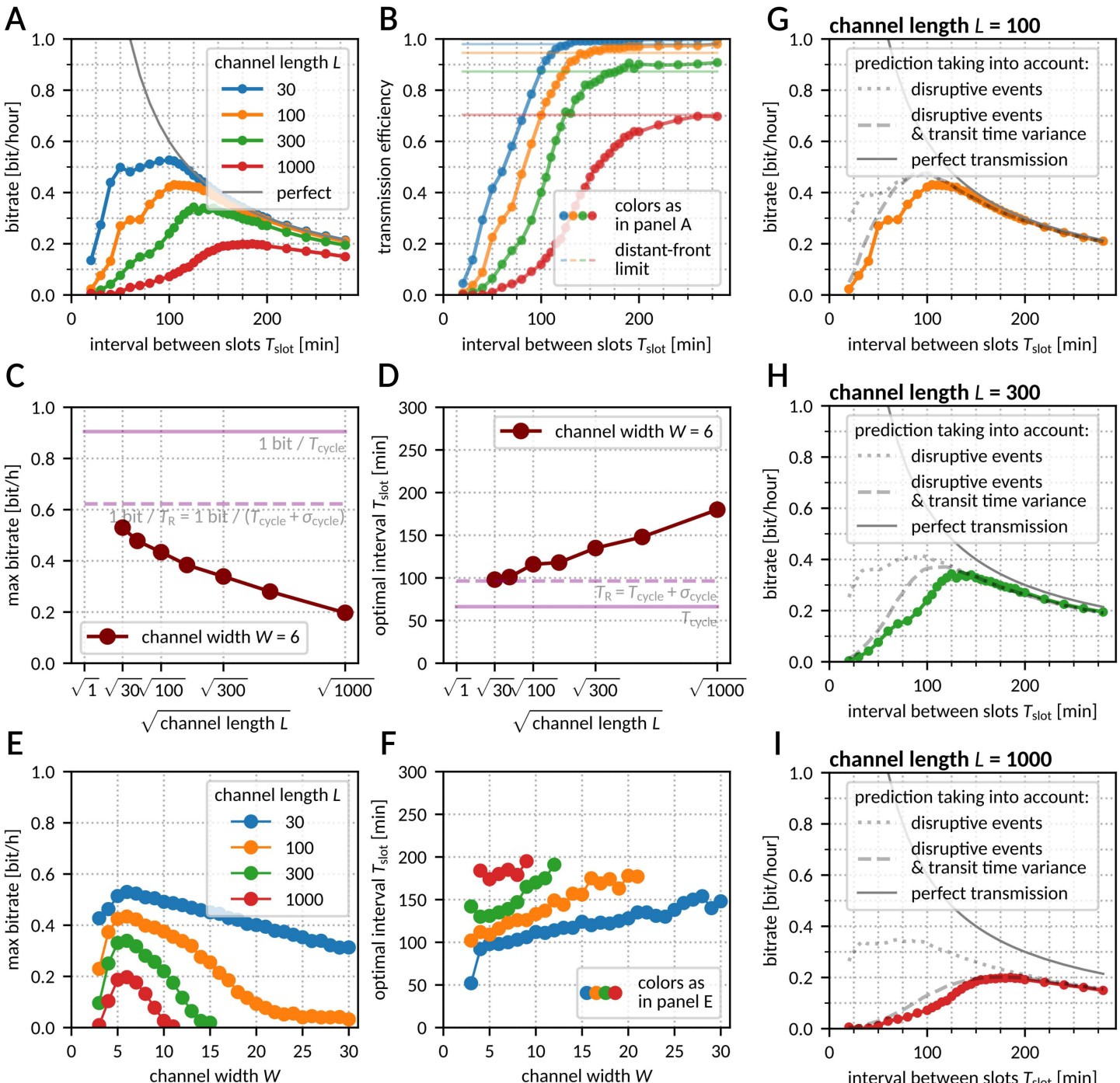

**Fig 4. Rate of information transmission. A** Bitrate as a function of the interval between slots for various channel lengths. The sent information rate is plotted in gray. Channel width $W = 6$. **B** Bitrate from panel A as a fraction of the sent information rate. Solid lines show the information transmission efficiency computed based on distant-front simulations, with a correction for backward fronts hitting the channel beginning (see Methods for details). **C, D** Maximum bitrate (C) and optimal inter-slot interval (D) as a function of (the square root of) the channel length. Channel width $W = 6$. **E, F** Maximal bitrate (E) and optimal inter-slot interval (F) as a function of the channel width for various channel lengths. **G–I** Prediction of bitrate taking into account the probability of front extinction due to propagation failure or collision with a backward front (dotted line), and the chance that a front was attributed to a wrong slot due to transit time stochasticity (dashed lines) – see Methods for details. Colored lines as in panel A. Data for panels A–B and G–I from $N = 100$ simulations with 500 front slots for each data point. Data for panels C–F was computed by searching for the maximum in a series of values computed as in panel A.

likely for longer channels. Once this effect is taken into account (see Methods for details), the prediction (Fig 4G–4I, dashed line) becomes satisfactory both for short and long $T_{slot}$, with some discrepancy for intermediate $T_{slot}$ values due to fronts spawned forward and other neglected factors.

In all the analyses, for sake of simplicity we employed the binary encoding protocol with equiprobable input symbols ($q = \frac{1}{2}$) and varying $T_{slot}$. As discussed in Text A in S1 Appendix, in cases where the effective refractory time $T_R$ is large compared to $\sigma_{transit}$ (e.g., for short channels or long $\tau_R$), the optimal $q$ can be substantially smaller than $\frac{1}{2}$, and consequently the maximum bitrate can be higher. However, only for $L = 30$ the bitrate increase was found significant (+20%).

Thus far, the bitrate was estimated based on the arrival times of individual fronts. We show and discuss the inference based on two consecutive fronts, which yields higher bitrate estimates for short channels ($L = 30$), in Text B in S1 Appendix.

## Sensitivity analysis

### Disruptive events in single fronts

The maximum front frequency and information transmission rate depend on model parameters. In Fig 5A we analyze how the kinetic model parameters influence the propensities of disruptive events, $\lambda_{fail}$ and $\lambda_{spawn}$, given in Eqs (1–2). For the nominal parameter values, $W_{fail} \approx 0$ and $W_{spawn} \approx 1$, and numerical analysis indicates that these coefficients remain in the range $(-1, 2)$ for the considered range of model parameters (Fig F in S1 Appendix). Thus, changes in these coefficients have modest effect on $\lambda_{fail}$ and $\lambda_{spawn}$. Therefore, crucial for understanding the system's behavior are the changes of coefficients $a_{fail}$ and $a_{spawn}$. It is important to notice that $a_{fail}$ changes approximately linearly with the kinetic parameters, while $a_{spawn}$ scales exponentially; as a consequence, both $\lambda_{fail}$ and $\lambda_{spawn}$ exhibit exponential dependence on $\tau_E$, $\tau_I$, $\tau_R$, and $\tau_{act}$.

We may notice that $a_{fail}$ (and thus $\lambda_{fail}$) does not depend on $\tau_E$ and $\tau_R$, as those parameters have no influence on whether a cell will be activated. The failure propensity increases with $\tau_{act}$ and decreases with $\tau_I$, because increase of the ratio $\tau_{act}/\tau_I$ implies a lower probability that an I cell activates a Q neighbor before proceeding to R.

The coefficient $a_{spawn}$ (and thus $\lambda_{spawn}$) depends on all four kinetic parameters. It increases with $\tau_{act}$, which can be explained as follows: high $\tau_{act}$ implies low cell activation propensity, which renders some cells activated at the rear side of the front and mediating front spawning. Importantly, the value of coefficient $a_{spawn}$ increases with $\tau_E$ (the increase of which implies a higher chance that the cell becomes inducible in the front tail) and decreases with $\tau_R$ (the increase of which implies a broader zone of R cells behind the front). Coefficient $a_{spawn}$ decreases with $n_E$, $n_I$, and $n_R$, because the increase of the number of intermediate states renders the distribution of times of transitions E → I, I → R, and R → Q narrower, resulting in less stochastic front propagation. Consequently, the minimum value of $\lambda_{tot}$ is an increasing function of $\tau_E$, decreasing function of $\tau_R$, and decreasing function of $n_E$, $n_I$, and $n_R$ (Fig 5B). Another consequence of the discussed dependence of $a_{spawn}$ on $\tau_E$ and $\tau_R$, and independence of $a_{fail}$ of these two kinetic coefficients, is that an increase of $\tau_R$ and/or decrease of $\tau_E$ shift $W_{opt}$ to higher values (Fig 5C).

In summary, we showed that an increase of the $\tau_R/\tau_E$ ratio (i.e., the ratio of the residence time in the refractory state to the residence time in the excited state) nearly exponentially reduces the disruptive event propensity $\lambda_{tot}$, and linearly increases the optimal width $W_{opt}$ of the channel.

### Periodic fronts and bitrate

Understanding how the model kinetic parameters influence the propensities of disruptive events allows us to analyze and interpret their impact on $T_{trans-min}$ and maximum bitrate (Fig 6). The

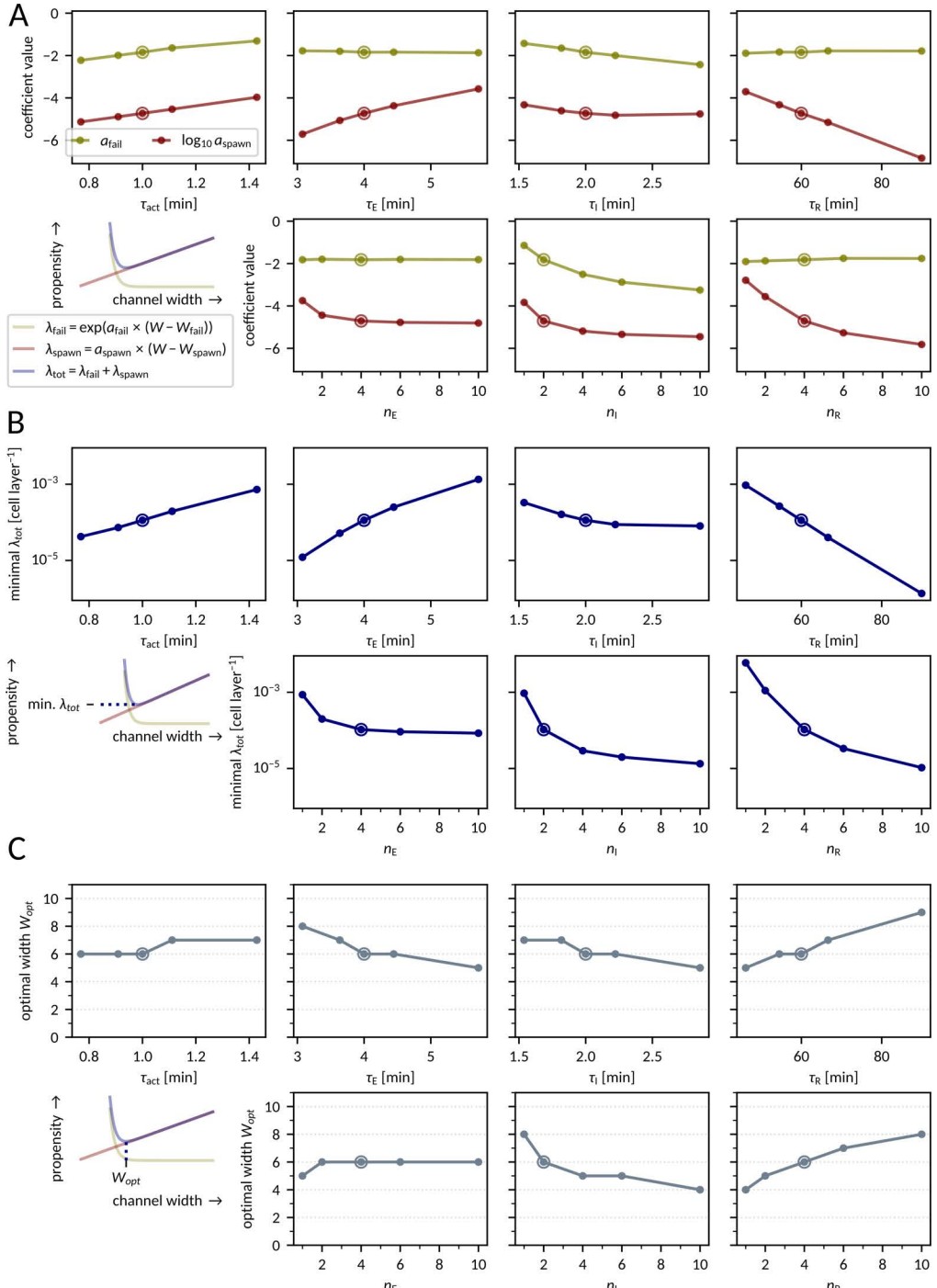

**Fig 5. Sensitivity analysis – disruptive events in single fronts. A** Dependence of slope coefficients $a_{spawn}$ and $a_{fail}$ on model parameters. **B** Dependence of the minimum total disruptive event propensity $\lambda_{tot}$ on model parameters. **C** Dependence of the optimal channel width $W_{opt}$ on model parameters. Each data point was obtained by fitting $\lambda_{spawn}(W)$ and $\lambda_{fail}(W)$ curves to numerical data as in Fig 2C. The encircled dots in each panel correspond to the nominal parameter set given in Fig 1B. All parameters apart from those indicated on the horizontal axis are fixed to their nominal values.

dependence of $T_{\text{trans-min}}$ on the on the $\tau_{\text{act}}$ and $\tau_{\text{I}}$ is dictated by the dependence of $\lambda_{\text{fail}}$ and $\lambda_{\text{spawn}}$ on these parameters; as expected, the increase of $\lambda_{\text{fail}}$ or $\lambda_{\text{spawn}}$ leads to the increase of $T_{\text{trans-min}}$. The dependence of $T_{\text{trans-min}}$ on $\tau_{\text{E}}$ and $\tau_{\text{R}}$ is more intriguing: $T_{\text{trans-min}}$ has maxima with respect to $\tau_{\text{E}}$ and decreases for large $\tau_{\text{E}}$; $T_{\text{trans-min}}$ decreases also for small $\tau_{\text{R}}$. This is puzzling, as we know from Fig 5A that large large $\tau_{\text{E}}$ and small $\tau_{\text{R}}$ imply large $\lambda_{\text{spawn}}$, which could block front propagation. However, as we can see in kymographs in Fig 6B and 6C, for small $\tau_{\text{R}}$ and large $\tau_{\text{E}}$ we observe a distinct pattern of front spawning. In this mode, numerous backward and forward fronts are created, so the front density (especially for small $\tau_{\text{R}}$) at the end of the channel is higher than at its beginning. In this regime it is impossible to coordinate collective cell motion in the direction of the front source.

Kymographs in Fig 6B also suggest that there should exist some optimal $\tau_{\text{R}}$, at which the bitrate is highest. In fact, this maximum is attained close to the nominal value of $\tau_{\text{R}}$, which is 60 min (Fig 6D). Larger values of $\tau_{\text{R}}$ do not allow for frequent fronts, while smaller ones are associated with a spawning front propagation pattern (blocking any information transmission). For the same reason, information transmission is blocked for large values of $\tau_{\text{E}}$. The information transmission rate increases monotonically with decreasing $\tau_{\text{E}}$. This is because the decrease of $\tau_{\text{E}}$ reduces $\lambda_{\text{spawn}}$ and to some extent the refractory time, and does not influence $\lambda_{\text{fail}}$.

Unsurprisingly, in the MAPK/ERK pathway, $\tau_{\text{E}}$ is short (several minutes) despite ERK activation being a multistep process. As shown in Fig G in S1 Appendix, the maximum bitrate grows monotonically with the number of sub-states (providing that the total time of all sub-states remains constant). In agreement with the influence of $\tau_{\text{act}}$ on $\lambda_{\text{fail}}$ and $\lambda_{\text{spawn}}$, the bitrate decreases monotonically with $\tau_{\text{act}}$, and, similarly as with $\tau_{\text{E}}$, $\tau_{\text{act}}$ appears to be short for the MAPK/ERK pathway. Finally, the bitrate attains its maximum close to the nominal value of $\tau_{\text{I}}$.

## Front propagation in broad channels

In broad channels of $W \gtrsim 15$, front spawning events frequently seed multiple fronts (Fig 2D and 2E) giving rise to chaotic front propagation patterns (as shown in Fig B in S1 Appendix and S4 Video), which can block information transmission for very long times. Thus, for such channels information transmission rate is close to zero, and one should rather ask about the number of fronts that can be transmitted before chaotic patterns develop due to spawning event(s). For this reason, we determined the expected number of fronts $N_{\text{fronts}}$ that can propagate through the channel of $L = 300$ and $W = 6, 15, 30$, and $60$ before the first spawning event. In Fig 7, lines correspond to noninteracting fronts ($\lambda_{\text{spawn}}{}^{-1}$ computed based on the $a_{\text{spawn}}$ and $W_{\text{spawn}}$ estimation shown in Fig 5A and Fig F in S1 Appendix), while circles were obtained from simulations with two fronts separated by the interval of $4\,\tau_{\text{R}}$. One can see that (for $L = 300$) such interval is sufficient to neglect the influence of inter-front interaction on spawning.

$N = 3000$ simulations were performed for each data point. Encircled points correspond to the nominal parameters given in Fig 1B.

We may notice that $N_{\text{fronts}}$ is a nearly exponentially increasing function of $\tau_{\text{R}}$ (Fig 7A) and a nearly exponentially decreasing function of $\tau_{\text{R}}$ (Fig 7B). Therefore, although $N_{\text{fronts}}$ decreases with the channel width as $1/W$ (for the nominal parameters, $N_{\text{fronts}}$ is about 10 times higher for $W = 6$ than for $W = 60$), the effect can be compensated with a relatively small change in the kinetic parameters. For example, for $W = 60$, increasing the refractory time to $\tau_{\text{R}} = 90$ min results in the increase of $N_{\text{fronts}}$ about 50 times, to over 100. Because for $W > 10$ the propensity of front propagation failure is negligible (Fig 2C), all initiated fronts will reach the end of the channel. Additionally, the transit time variance $\sigma^2_{\text{transit}}$ is a decreasing function of $W$ (Fig A panel b in S1 Appendix), so for $W > 10$, $L = 300$ the distribution of the transit time is also

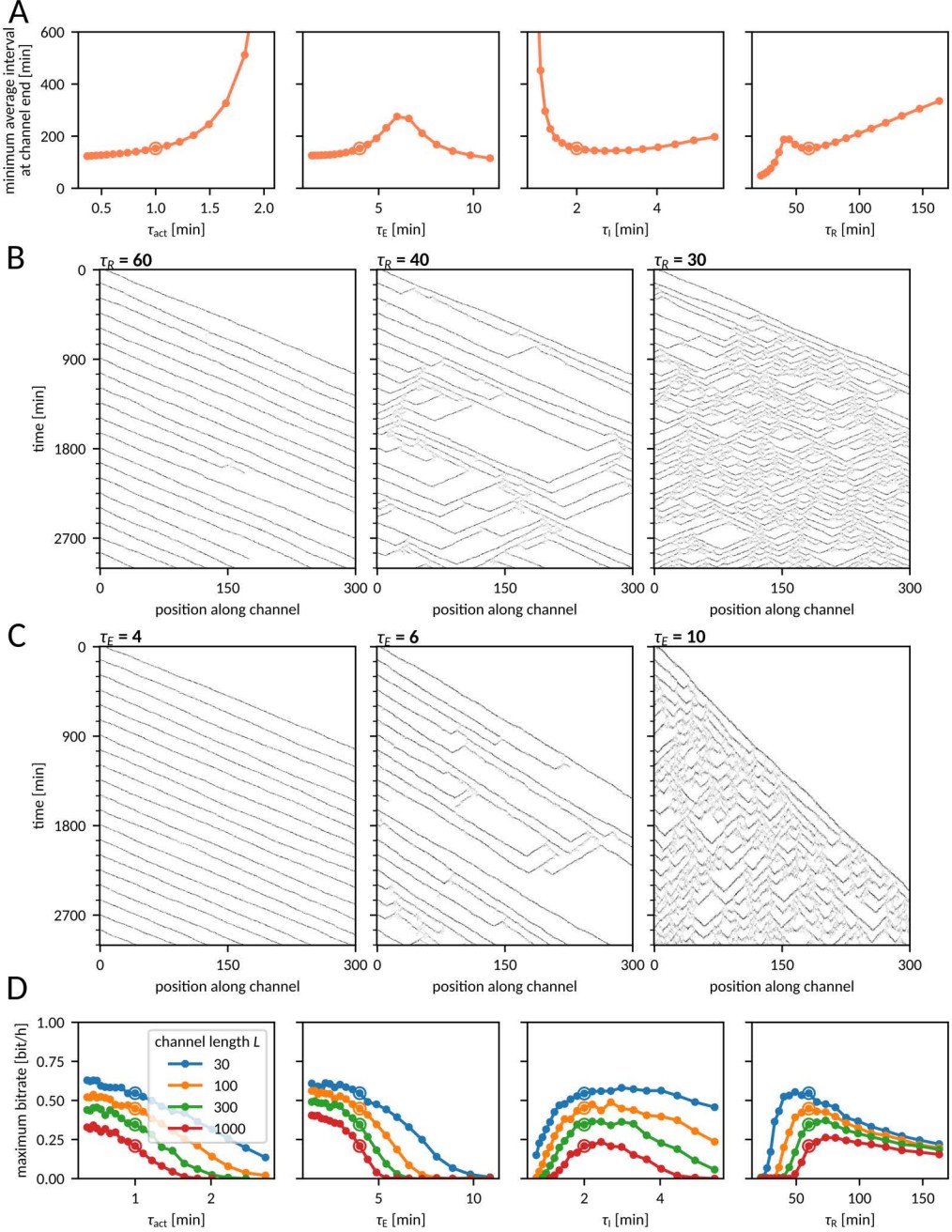

**Fig 6. Sensitivity analysis – periodic fronts and the bitrate. A** Interval $T_{\text{trans-min}}$ (minimized with respect to the initial interval) at the channel end of the channel as a function of the model parameters. The remaining parameters are equal to the nominal values given in Fig 1B. **B, C** Kymographs of the number of active cells (E or I) as a function of time and distance along the channel for fronts initiated periodically at the interval of 150 min for three different values of $\tau_R$ (B) and $\tau_E$ (C). The remaining parameters are equal to the nominal values given in Fig 1B. **D** Maximal bitrate (maximized with respect to the input bitrate) as a function of the model parameters. In all panels, channel dimensions are $W = 6$ and $L = 300$. Encircled dots correspond to the nominal parameters given in Fig 1B.

sufficiently narrow for the received fronts to be uniquely assigned to their slots (for $T_{\text{slot}} \geq 4\tau_R$). Therefore, for the aforementioned example parameters ($L = 300$, $W = 60$, $\tau_R = 90$ min, and other parameters having nominal values), before the channel is blocked, on average

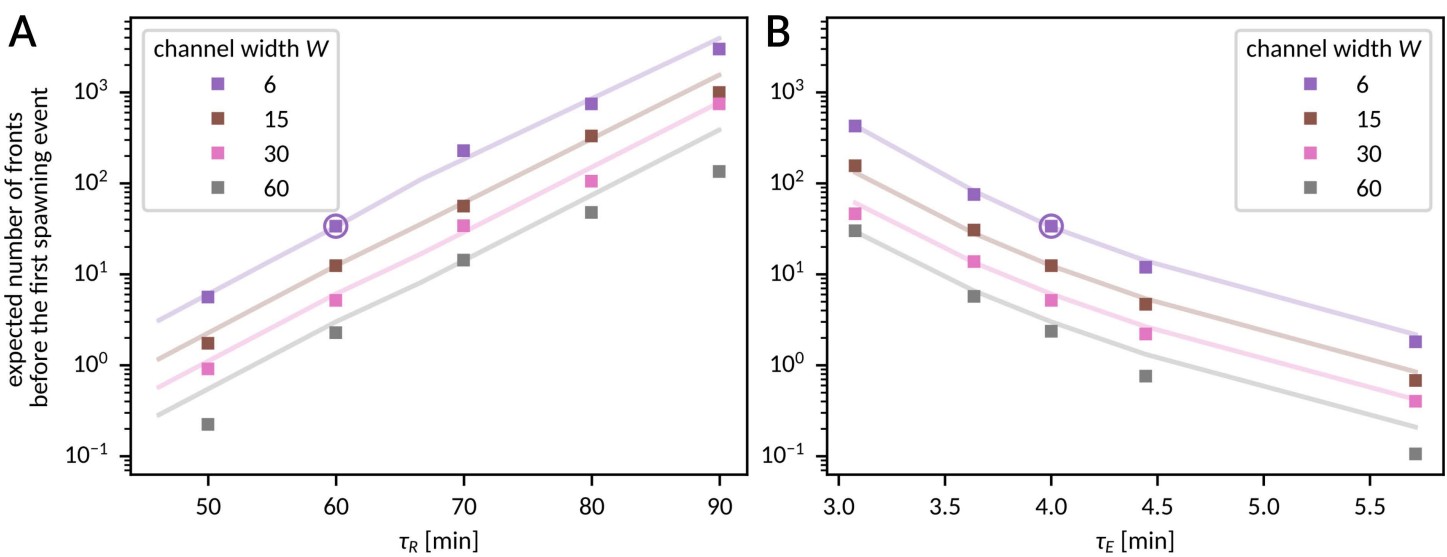

**Fig 7. Sensitivity analysis – front propagation in broad channels. A, B** Expected number of fronts that will pass uninterruptedly through the channel of length $L = 300$ before the first spawning event as a function of the model parameters $\tau_R$ (A) and $\tau_E$ (B). Computed based on the spawning probability for the latter of two fronts initiated $4 \times \tau_R$ apart (squares) or based on the coefficients $a_{spawn}$, $W_{spawn}$ from Fig 5A and Fig F in S1 Appendix (lines).

$N_{fronts} \approx 100$ can be sent, received, and properly assigned to their slots, which allows for transmitting 200 bits of information in the binary protocol with equiprobable symbols.

## Discussion

ERK activation triggers cell contraction, which leads to activation of EGFR in neighboring cells [3]. In cell collectives, this mechanochemical coupling coordinates the propagation of waves of ERK activity and cell movement against the waves' direction [9]. As the MAPK/ERK cascade is inhibited behind the front, cell contraction induces subsequent ERK activation in cells directly ahead of the front, rather than behind it. The mechanochemical coupling was theoretically studied in 1-D and 2-D models by Boocock et al. [4,17]. Their model, constrained with data obtained from experiments on MDCK cells but omitting details of signal transduction through the MAPK/ERK cascade, allowed them to determine the optimal wavelength and period for maximizing migration speed towards the tissue boundary. In our study, we investigated the processes that interfere with the stochastic propagation of activity waves.

Canonically, trigger (homoclinic) traveling waves employ a positive feedback to propagate over long distances [18–20]. In our model, the positive feedback (at the tissue level) arises when the inducing cell excites a quiescent cell, and then the quiescent cell becomes inducing itself. Because after the passage of a homoclinic wave, the system returns to its single steady state, such waves can be initiated recurrently at desired time points; the same property entails that in the presence of stochastic fluctuations, waves can vanish but also may arise spontaneously.

In our study, we characterized two types of disruptive events: front propagation failure and new front spawning. Propagation failure eliminates the front, whereas a single backward-spawned front collides and annihilates with the subsequent front in the series. Thus, when fronts are initiated repeatedly, in both cases, the total number of fronts is reduced by 1. Consequently, the probability that a front in a series of fronts passes through the channel decreases with the total disruptive event propensity $\lambda_{tot} = \lambda_{fail} + \lambda_{spawn}$. Importantly, because $\lambda_{fail}$ decreases (exponentially), while $\lambda_{spawn}$ increases (linearly) with the channel width $W$, there is some

optimal channel width $W_{opt}$, for which the probability of uninterrupted front propagation through the channel is the highest. This result is surprising because intuitively, the reliability of stochastic signal transduction should monotonically grow with the channel width. The fidelity of front propagation in relatively narrow channels may allow for transmission of ERK waves in narrow structures like capillary and lymphatic vessels, in which sequential ERK activation [21] and calcium waves [22] have been observed.

The tail of each front consists of a layer of refractory cells, which block the propagation of other fronts close behind. The effective refractory time $T_R = T_{cycle} + \sigma_{cycle}$, where $T_{cycle}$ is the cell cycle time (from Q through E, I, and R back to Q) and $\sigma_{cycle}$ is the cell cycle standard deviation. When studying periodically initiated fronts, we observed that the time $T_R$ approximates the lower limit on the time interval between fronts that can be transmitted through a short channel. In longer channels, the minimum average interval between fronts $T_{trans-min}$ is larger than $T_R$, and increases with the channel length.

To numerically estimate the rate at which information can be transmitted, we employed a binary encoding protocol. We found that for the optimal channel width, the bitrate is 0.5 bit/hour for $L = 30$ and 0.2 bit/hour for $L = 1000$ for the nominal model parameter values, consistent with the timescales of processes implicated in the MAPK/ERK signaling cascade. There are two major phenomena that limit the information transmission rate: disruptive events and transit time stochasticity. For long intervals between fronts, the possibility of front extinction due to disruptive events is the limiting factor. In this case, we showed that the amount of information transmitted per slot equals $MI_{slot} = 1 - 1/2 \times [(p+1) \log_2 (p+1) - p \log_2 p]$, where $p$ is the probability of front extinction, which is an increasing function of front slot frequency. Since the bitrate is the product of the slot frequency and $MI_{slot}$, it attains its maximum for some optimal inter-slot interval. When the interval between front slots is comparable to or shorter than the standard deviation of the transit time (that increases with channel length), fronts reaching the end of the channel may be attributed to a wrong slot. Once this effect is taken into account, our phenomenological predictions become satisfactory for both short and long inter-slot intervals. As expected, the bitrate reaches its maximum for the channel width which minimizes the propensity of disruptive events.

We performed a sensitivity analysis to show that the ratio of the refractory to the excited state residence times ($\tau_R/\tau_E$) nearly exponentially reduces the total disruptive event propensity. Surprisingly, we found that $T_{trans-min}$ decreases for large $\tau_E$ and small $\tau_R$, attaining high values when the disruptive event propensity is high. Kymographs indicate that for small $\tau_R$ or large $\tau_E$ there is a distinct chaotic front spawning regime. In this regime, multiple backward and forward fronts are created, so that the front density (especially for small $\tau_R$) at the end of the channel is higher than at its beginning. The existence of multiple backward fronts excludes coordination of collective cell motion in the direction of the signaling source. We found that (for other parameters fixed) there exists an optimal $\tau_R$ (close to the nominal $\tau_R$ value of 60 min) associated with the highest bitrate. Larger $\tau_R$ precludes frequent fronts, while smaller $\tau_R$ leads to the chaotic front spawning regime blocking any information transmission. Information transmission is also blocked for large values of $\tau_E$, and bitrate increases monotonically with decreasing $\tau_E$. This is because a decrease of $\tau_E$ reduces $\lambda_{spawn}$ and does not influence $\lambda_{fail}$. Unsurprisingly, in the MAPK/ERK pathway, $\tau_E$ is short despite ERK activation being a multi-step process.

Finally, we observed that in broader channels, front spawning causes that front propagation becomes chaotic, as observed in the experimental work of Hino et al. [3]. However, sensitivity analysis indicates that front spawning propensity grows only linearly with the channel width while it decreases exponentially with the refractory time. Therefore, a relatively modest increase of the refractory time allows for unperturbed transmission of multiple fronts

also in broader channels before chaotic front patterns develop. This may explain why the ERK activity waves observed in zebrafish scales [5] are much more stable.

Shannon mutual information and information rate give an upper bound on the complexity of decisions that the receiving agent can make in response to the sent signal. It was recently demonstrated that the navigation of *Escherichia coli* in changing chemoattractant gradients is information-limited [23]. For a cell to informedly choose among two possible responses, it has to receive at least $\log_2 2 = 1$ bit of information. If a cell is to make such binary decisions every ten minutes, the required information rate is 6 bit/h. Previous studies have established that the NF-κB and MAPK/ERK pathways are able to transmit merely 1 bit of information about the strength of a stimulus enabling binary decision-making [24,25]. In a recent work, we have established that the MAPK/ERK cascade is capable of transmitting information between membrane (opto)receptor (opto-FGFR) and ERK at bitrate exceeding 6 bit/hour [26]. This high bitrate allows for coordination of fast processes, such as mitotic divisions in *Drosophila melanogaster*, that occur synchronously about every 10 min [27].

To coordinate behavior of a cell population, as in the case of wound healing, information must be propagated between cells, and the bitrate limits temporal resolution, complexity of a desired behavior, and the extent of cell coordination. Travelling fronts (in contrast to diffusion) can propagate signals at a constant speed over long distances. Our analysis of ERK activity waves in cell monolayers indicates that there is an optimal channel width and optimal slot frequency, at which the front propagation is least perturbed and consequently the highest bitrate is achieved. Additionally, sensitivity analysis suggests that the time scales associated with signal transmission within the MAPK/ERK pathway are in the range that allows for efficient information transmission by traveling ERK activity waves.

## Methods

### Numerical simulations

Kinetic Monte Carlo simulations on a triangular lattice were carried out according to the Gillespie algorithm using the code adapted from Ref. [28]. In all simulations except those for Fig C in S1 Appendix, periodic boundary conditions were applied along the lower and the upper edges. Cells traverse the QEIRQ sequence of (sub)states when activated by a neighboring cell in an I substate. Cells cannot change their locations nor die. State-to-state transitions are shown in Fig 1A and default parameter values are given in Fig 1B.

### Front tracking

**Front arrival time.** Activity (i.e., total number of cells in states E or I, $N_{EI}$) in the last cell layer was counted as a function of time; local maxima along the temporal axis were found and filtered based on peak height to remove peaks closer than 20 or 30 min apart.

**Disruptive event detection.** Activity (i.e., the total number of cells in states E or I, $N_{EI}$) was counted as a function of the distance $x$ from the beginning of the channel and time to form a kymograph ($x$–$t$ plot; Figs 3A, 6B, and 6C). To determine front positions, the kymograph was smoothed (with a Gaussian kernel along the $x$-axis and an exponential kernel along the $t$-axis); then local maxima of $N_{EI}$ along the $x$-axis were found (thresholds on minimal peak height and minimal distance between peaks were used to discard short-lasting fluctuations). Fronts were tracked using LapTrack [29]. Fates of fronts that did not reach the end of the channel were classified as 'annihilation', 'immediate failure', or 'propagation failure' depending on whether, respectively, the front disappeared in the vicinity of another front propagating in the opposite direction, within a certain minimal distance from the channel beginning (1–5 cell layers depending on the channel width), or while propagating freely through the channel. To detect

front spawning, track splits were recorded. Tracks shorter than 50 min that did not reach the channel end nor split into other tracks were discarded together with the split in which they were created. The remaining track splits were treated as front spawning and grouped into events: two splits were assigned to a single spawning event if they occurred not further than 20 cell layers apart in space and 150 min away in time (Fig 2D and 2E).

## Numerical data analysis

**Front speed and transit time variance.** To determine the average inverse front speed and transit time variance (Fig A in S1 Appendix), for each channel width, $N = 30,000$ simulations with a single front were performed in a reactor of length $L = 300$ (or $L = 30$ in the cases when, due to a very short average front range, too few fronts reached the distance of 300 to collect sufficient statistics). Average inverse front speed and transit time variance were computed as $\langle v^{-1} \rangle = \langle \tau_{transit} \rangle / L$ and $\sigma^2_0 = (\text{Var } \tau_{transit}) / L$, where $\tau_{transit}$ is the time of the first detected front arrival in each simulation. Fronts that did not reach the end of the channel were discarded; therefore, formally, we compute both quantities conditioned on the fact that the front reached the channel end.

**Disruptive events – single fronts.** For each channel width, $N = 30,000$ simulations with a single front were performed in a reactor of length $L = 300$; fronts were tracked, and front fates were determined. For each simulation, the first disruptive event (propagation failure or front spawning) was determined and its position was recorded. The number of simulations with at least one disruptive event, $n_{event}$, and the average position of the first event, $x_{event}$, were obtained. The total event propensity was calculated with the formula

$$\lambda_{tot} = \frac{n_{event}}{n_{event} \times x_{event} + (N - n_{event}) \times L}. \tag{4}$$

Propensities of events of a particular type were computed as

$$\lambda_{fail} = \frac{n_{fail}}{n_{event}} \lambda_{tot}, \tag{5}$$

$$\lambda_{spawn} = \frac{n_{spawn}}{n_{event}} \lambda_{tot}, \tag{6}$$

where $n_{fail}$ and $n_{spawn}$ denote the number of simulations in which, respectively, propagation failure and front spawning was the first disruptive event. We used the least squares method for fitting coefficients $a_{spawn}$, $a_{fail}$, $W_{spawn}$, $W_{fail}$ in Eqs (1–2) (Figs 2C, 5A, and Fig F in S1 Appendix). The optimal width $W_{opt}$ in Fig 5C was taken as the integer with minimal $\lambda_{tot}$, computed according to Eqs (1–3) based on the best-fit coefficients. Minimal disruptive event propensity (Figs 5B and 6A) was computed using Eqs (1–3) for $W = W_{opt}$. The estimation of the number of fronts spawned in either direction (Fig 2D and 2E) was based on the first spawning event in each simulation.

**Disruptive events – interacting fronts.** For each considered inter-front interval, $N = 3000$ simulations were performed with two fronts separated by the given (initial) interval; fronts were tracked and their fates were determined. Based on this, the probability of disruptive events concerning the latter front (failure, spawning, or annihilation by a front spawned by the first front) was calculated (Fig 3B).

**Front arrival frequency.** The average frequency of fronts at the end of the channel (Fig 3C and 3E) was measured by counting front arrivals at the end of the channel of indicated length.

The frequency of fronts reaching a certain distance (Fig 3D) was measured by counting peaks of activity along the temporal axis at the given distance from the beginning of the channel, in a channel of length $L = 1000$. $N = 30$ simulations with 500 fronts each were performed for each initial inter-front interval and (for Fig 3C and 3E) channel length.

**Information transmission rate – numerical estimation.** To estimate the information transmission rate (Figs 4, and 6D; Fig G, H , panel b, and I in S1 Appendix), $N = 100$ random sequences of 500 binary symbols $S$ were generated, determining, for each slot, whether a front should be initiated. Unless stated otherwise, the probability of initiating a front was ½. For each channel width, the average inverse front velocity was estimated numerically as described above (based on $N = 3000$ simulations) and used to predict the expected front transit time. For each front sequence, simulations were performed and, for each slot (with associated time $t_{\mathrm{slot}}$), we computed the expected arrival time $t_{\mathrm{expected}} = t_{\mathrm{slot}} + \langle \tau_{\mathrm{transit}} \rangle = t_{\mathrm{slot}} + L \langle v^{-1} \rangle$ and recorded the difference $\Delta t = t_{\mathrm{arrival}} - t_{\mathrm{expected}}$ between the closest arrival time ($t_{\mathrm{arrival}}$) and $t_{\mathrm{expected}}$. Note that we computed $t_{\mathrm{expected}}$ and $\Delta t$ for each slot, regardless of whether a front was initiated in it or not. In Fig E in S1 Appendix we show histograms of $\Delta t$ for $S = 1$ and $S = 0$.

Next, we calculated the mutual information per slot $\mathrm{MI}_{\mathrm{slot}}$ as information between the binary symbol $S$ and variable $\Delta t$ as follows. The differences $\Delta t$, binned with 1-minute resolution, were counted across all slots and stimulation sequences. Conditional entropy $\mathrm{H}(S \,|\, \Delta t)$ was computed using a $k$NN-based algorithm with Miller–Madow bias correction [30,31]. In short: if a particular difference $\Delta t'$ occurred $n \geq k = 25$ times, numbers of occurrences originating from slots with $S = 0$ and $S = 1$ were counted ($n_0$ and $n_1$, respectively) and the following formula was used for conditional entropy estimation:

$$\mathrm{H}\left(S \,|\, \Delta t = \Delta t'\right) = \frac{n_0}{n}\log_2\frac{n_0}{n} + \frac{n_1}{n}\log_2\frac{n_1}{n} + \mathrm{MM}, \tag{7}$$

where $\mathrm{MM} = 1/(2n \ln 2)$ if $n_0 \neq 0$ and $n_1 \neq 0$, and $\mathrm{MM} = 0$ otherwise. If $n < k$, in order to ensure proper sample size, the calculation of $n_0$, $n_1$ and $n$ was repeated including not only points with $\Delta t = \Delta t'$, but also with $\Delta t$ closest to $\Delta t'$ so that $k$ data points were used in total.

The conditional entropy $\mathrm{H}(S \,|\, \Delta t = \Delta t')$ was averaged across all data points to obtain $\mathrm{H}(S \,|\, \Delta t)$, and subtracted from $\mathrm{H}(S) = 1$ bit to obtain $\mathrm{MI}_{\mathrm{slot}}$ according to the formula

$$\mathrm{MI}_{\mathrm{slot}} = \mathrm{I}(S; \Delta t) = \mathrm{H}(S) - \mathrm{H}(S \,|\, \Delta t). \tag{8}$$

In cases where the $\mathrm{MI}_{\mathrm{slot}}$ estimation was based on more than one front (Text A and Fig I in S1 Appendix), a vector of arrival time differences with respect to the expected arrival time was used in place of $\Delta t$; Euclidean norm was used to find the nearest neighbors in the case of $n < k$.

Bitrate optimization was conducted by scanning the parameter space around a value predicted by earlier results (Figs 4E, 4F, 6D, and Fig G in S1 Appendix) or by a custom algorithm (Fig 4C and 4D).

**Information transmission rate – semi-analytical predictions.** To predict the information transmission rate for distant fronts (Fig 4B), we used the coefficients $a_{\mathrm{spawn}}$, $a_{\mathrm{fail}}$, $W_{\mathrm{spawn}}$, and $W_{\mathrm{fail}}$ obtained based on single-front simulations (Fig 2C; Eqs (1–2)) to calculate $\lambda_{\mathrm{spawn}}$ and $\lambda_{\mathrm{fail}}$. Then, we computed the probability that a front in a front train is eliminated

$$p = \exp\left(-L \times \left(\lambda_{\mathrm{fail}} + \gamma \times \lambda_{\mathrm{spawn}} \times n_{\mathrm{backward}}\right)\right), \tag{9}$$

where $n_{\mathrm{backward}} = 1.285$ is the expected number of backward fronts spawned in a single event (obtained from data shown in Fig 2D) and $\gamma$ is the probability that a backward front collides with a next front before reaching the channel beginning. We used $\gamma = 1$ to obtain the

distant-front value shown in Fig 4B. Based on the probability of front extinction $p$ we constructed the confusion matrix

$$
\begin{array}{c|cc}
\text{sent} \setminus \text{received} & 0 & 1 \\
0 & 1-q & 0 \\
1 & q\,p & q(1-p)
\end{array}
\quad, \tag{10}
$$

where $q$ is the probability of sending '1' in the binary protocol. Then we computed $MI_{slot}$ using the standard formula

$$
MI_{slot} = \sum_{i,j} c_{ij} \left( \log_2 c_{ij} - \log_2 \sum_{i'} c_{i'j} - \log_2 \sum_{j'} c_{ij'} \right) \tag{11}
$$

where $c_{ij}$ are the entries of the confusion matrix, obtaining

$$
MI_{slot} = -q\,\log_2 q \;+\; (1-q+pq)\log_2(1-q+pq) \;+\; pq\log_2(pq), \tag{12}
$$

which in the case of $q = \frac{1}{2}$ simplifies to

$$
MI_{slot} = 1 - \frac{1}{2}\left( (p+1)\log_2(p+1) - p\log_2 p \right). \tag{13}
$$

Eventually, we computed the information transmission rate as

$$
r = \frac{MI_{slot}}{T_{slot}}. \tag{14}
$$

To include the effect of interaction between fronts (Fig 4G–4I, dotted line), we took into account the probabilities of disruptive events $p_{fail} = p_{propagation\ failure} + p_{immediate\ failure}$ and $p_{spawn}$ obtained from the numerical simulations with two fronts at different intervals (Fig 3B), as a function of the initial interval. To account for the fact that in the binary protocol the initial interval between fronts can be any multiplicity of the inter-slot interval $T_{slot}$ (interval of length $k \times T_{slot}$ has probability $q(1-q)^{k-1}$), we averaged the probabilities $p_{fail}$ and $p_{spawn}$ using the formula

$$
p_{event} = \sum_{k=1}^{\infty} q(1-q)^{k-1} \times p_{event}(k \times T_{slot}), \tag{15}
$$

where $p_{event}$ is either $p_{fail}$ or $p_{spawn}$ and $q = \frac{1}{2}$ is assumed. Then, we approximated the probability that a front is eliminated with the formula

$$
p = 1 - \frac{1 - p_{fail}}{1 \;+\; \gamma \,\times\, p_{spawn} \,\times\, n_{backward}}, \tag{16}
$$

where $n_{backward} = 1.285$ was taken from the single-front simulations and $\gamma$ was calculated as

$$
\gamma = \sum_{k=1}^{\infty} q(1-q)^{k-1} \times \max\left( 0; \; 1 - k \times \frac{v\, T_{slot}}{2L} \right). \tag{17}
$$

Based on $p$, we calculated $MI_{slot}$ and bitrate using Eq (12) and Eq (14). Note that, unlike before, in this approach we used probabilities of disruptive events rather than propensities, which required running the two-front simulations for each channel length separately (analogous to Fig 3B, in which only results for $L = 300$ are presented). This was necessary, as the probabilities do not scale linearly with the channel length, due to the fact that the disruptive events are more likely to take place near the beginning of the channel.

To additionally include the effect of the transit time variance (Fig 4G–4I, dashed line), we assumed that the transit time has a normal distribution with $\sigma^2 = \sigma^2_{transit} = \sigma^2_0 L$ and that a '1' was received whenever a front arrived within $T_{slot}/2$ from $t_{expected}$ (note that is *not* what we do in the numerical approach). The probability of a front being attributed to the correct slot, conditioned on that it reaches the channel end, is then $p_{accurate} = \mathrm{erf}(T_{slot} / (2\sqrt{2}\,\sigma_0\sqrt{L}))$. We modified the elimination probability $p' = p + (1-p)(1-p_{accurate})$ and computed the probability that a front was mistakenly detected in a slot in which no front was initiated $p_{fake} = q(1-p)(1-p_{accurate}) - 1/4\, q^2(1-p)^2(1-p_{accurate})^2$ (the latter term prevents double counting of cases in which fronts from both the previous and the next slot are attributed to the considered slot). We created the confusion matrix:

$$
\begin{array}{c|cc}
\text{sent} \setminus \text{received} & 0 & 1 \\
0 & (1-q)(1-p_{fake}) & (1-q)\,p_{fake} \\
1 & qp'(1-p_{fake}) & q\big[(1-p') + p'p_{fake}\big]
\end{array}
\tag{18}
$$

and computed $\mathrm{MI}_{slot}$ using the formula (11).

## Supporting information

**S1 Appendix. Containing the following material** : **Fig A.** Front propagation speed. **Fig B.** Chaotic front spawning. **Fig C.** Propensity of disruptive events for inert boundary conditions. **Fig D.** Front speed reduction due to the vicinity of a previous front. **Fig E.** Histogram of the temporal distance $\Delta t$ from the expected arrival time to the nearest arrival time, conditioned on whether a front was initiated in the given slot. **Fig F.** Dependence of $W_{fail}$ and $W_{spawn}$ on model parameters. **Fig G.** Bitrate dependence on the number of E, I, and R substates. **Text A with Fig H.** Maximum bitrate for non-equiprobable binary symbols. **Text B with Fig I.** Alternative decoding: Bitrate estimation based on more than one front arrival time.
(PDF)

**S1 Video. Propagation failure.** Simulation time-lapse related to Fig 2A.
(MP4)

**S2 Video. New front spawning.** Simulation time-lapse related to Fig 2A.
(MP4)

**S3 Video. Front collision and annihilation.** Simulation time-lapse related to Fig 2B.
(MP4)

**S4 Video. Chaotic front spawning.** Simulation time-lapse related to Fig B in S1 Appendix
(MP4)

## Author contributions

**Conceptualization:** Paweł Nałęcz-Jawecki, Marek Kochańczyk, Tomasz Lipniacki.

**Formal analysis:** Paweł Nałęcz-Jawecki, Przemysław Szyc, Frederic Grabowski, Marek Kochańczyk.

**Funding acquisition:** Tomasz Lipniacki.

**Investigation:** Paweł Nałęcz-Jawecki, Przemysław Szyc, Frederic Grabowski, Marek Kochańczyk, Tomasz Lipniacki.

**Software:** Paweł Nałęcz-Jawecki, Przemysław Szyc, Frederic Grabowski, Marek Kochańczyk.

**Visualization:** Paweł Nałęcz-Jawecki, Przemysław Szyc, Marek Kochańczyk.

**Writing – original draft:** Paweł Nałęcz-Jawecki, Marek Kochańczyk, Tomasz Lipniacki.

**Writing – review & editing:** Paweł Nałęcz-Jawecki, Frederic Grabowski, Marek Kochańczyk, Tomasz Lipniacki.

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
