## [Decision Letter · Decision Letter 0]

16 Sep 2024

Dear Prof. Lipniacki,

Thank you very much for submitting your manuscript "Information transmission in a cell monolayer: A numerical study" for consideration at PLOS Computational Biology.

As with all papers reviewed by the journal, your manuscript was reviewed by members of the editorial board and by several independent reviewers. In light of the reviews (below this email), we would like to invite the resubmission of a significantly-revised version that takes into account the reviewers' comments. In particular, technical comments by the reviewers 1 and 3 deserve close attention. Moreover, the justification of biological relevance of information capacity calculations should be extended.

We cannot make any decision about publication until we have seen the revised manuscript and your response to the reviewers' comments. Your revised manuscript is also likely to be sent to reviewers for further evaluation.

Sincerely,

Oleg A Igoshin

Academic Editor

PLOS Computational Biology

Stacey Finley

Section Editor

PLOS Computational Biology

Reviewer's Responses to Questions

**Comments to the Authors:**

Reviewer #1: The authors investigate how information is transmitted through cell monolayers, motivated by earlier experimental observations of the spatiotemporal waves of MAPK/ERK activity in developing and regenerating tissues. Employing a stochastic cellular automaton framework, they evaluate the efficiency of long-range communication by analyzing the rate of information transmission within 2D rectangular cellular channels. The study identifies key factors that disrupt this process, including front propagation failure, the spawning of new fronts, and variability in front velocity. The authors highlight the importance of optimal channel width, which geometrically determines front length, as a critical factor in maximizing information transmission. Moreover, they demonstrate that the timescale of biochemical reactions in the MAPK cascade is critical to maximize information transmission rate in the tissues.

The modeling approach is unique for its ability to simplify the complex mechanistic processes underlying ERK activity propagation in tissues, thereby enabling an exploration of how timescales of cellular processes and the geometry of the field influence the success or failure of information transmission across cells. This simplicity, however, is a double-edged sword. While the model may not fully capture the intricacies of biological phenomena, its simplicity allows for a broader examination of the general principles governing the optimal design of biological systems for cell-to-cell information transfer. To fully leverage the strengths of their approach, the authors could have expanded the scope of their analysis and discussion beyond the current ones.

I believe this manuscript provides a unique and intriguing contribution that would appeal to the broad readership of this journal. However, as outlined below, there are several critical points where essential information is lacking, and the authors need to address the following points.

Here are some comments for suggested improvements:

1. The authors have compared the deterministic and stochastic regimes of the QEIRQ model, emphasizing on the stochastic regime. However, the manuscript lacks in detailing the implementation of stochastic fluctuations. In the Model section, it is mentioned that the time scales follow Erlang distributions, yet the specific parameter values are not provided. As Erlang distributions are defined by two parameters and can vary significantly depending on these values, the absence of this information makes it difficult to fairly evaluate the results and conclusions. It is essential for the authors to clarify the model, as without these details, it is impossible to fully understand the nature of the stochasticity they discuss or to accurately assess their findings.

2. Essential information regarding the numerical simulations, such as boundary and initial conditions, is missing from the Method section. Although the legend of Figure 1B mentions that 'Periodic boundary conditions are applied along the longer edge,' this should be explicitly described in the Method section if it applies to other simulations as well. Detailed information about the simulation settings is necessary to ensure clarity and reproducibility.

3. It appears that the authors have applied periodic boundary conditions on the upper and lower sides of the channels in their simulations. While this setting may be appropriate for modeling duct-like structures, it may not be suitable for certain biological tissues. It is crucial to explore how the results might change under different boundary conditions, as system's behavior under the current setting would highly depend on the bond connections between constituent cells or the medium in the domain. Performing this additional numerical investigation would significantly enhance the generality and robustness of the conclusions drawn.

4. The authors claim that the channel width of 6 is optimal; however, this conclusion appears to be highly sensitive to parameter values, particularly those related to stochasticity. To provide a clearer understanding, the authors should first specify the exact parameter sets they used (as also related to my comment #1) and then show how the optimal width varies with changes in the stochastic parameters."

5. In the last paragraph of subsection Numerical results, the manuscript simply states ‘The optimal inter-slot interval changes consistently with the maximum bitrate (Fig 4F)’. The authors should provide a more detailed explanation of the supporting evidence and data related to this point.

6. In the final paragraph of the Discussion section, the authors claim that ‘Additionally, sensitivity analysis suggests that the MAPK/ERK pathway is “optimized” for efficient information transmission by traveling ERK activity waves.’ However, the results presented in this study are based entirely on theoretical models. In addition, the current modeling framework is not fully aligned with the detailed biochemical and mechanical processes elucidated by previous experimental research. Given these limitations, the authors should consider moderating this claim to better reflect the theoretical nature of their findings.

7. In the developing or regenerating fins of zebrafish, as reported by De Simone et al. (Nature, 2021, doi: 10.1038/s41586-020-03085-8), concentric waves propagate unidirectionally in 2D space from a specific central point within the fins. Could the authors explain this phenomenon using their current model? How does the model assumption correspond with the ideas proposed in De Simone’s paper? Additionally, a theoretical paper by Hayden et al. (Biophys J, 2021, doi: 10.1016/j.bpj.2021.05.004) offers relevant insights from a theoretical viewpoint. It would benefit readers to include a discussion on whether and how the current model aligns with the model proposed in Hayden's work.

Minor points:

8. Figure 5C is not cited in the main text.

9. In the final paragraph of Results, the manuscript states 'the bitrate also grows monotonically with τ_act,' but should this actually be 'the bitrate also decreases monotonically with τ_act'? It looks so in the leftmost panel of Figure 6D. Please verify this.

10. In the second paragraph of Discussion section, the manuscript states ‘capable of transmitting information between membrane (opto)receptors and ERK’. Please explain what the membrane (opto)receptors indicate.

Reviewer #2: In their manuscript, the authors study how cell-cell communications allow propagation of fronts using a toy model of the ERK pathway. The authors semi-quantitatively describe the failures in front propagation using three modes of failure. Interestingly, the authors find that there is an optimal information transmission rate has a natural optimum. In a periodic setting, when pulses are sent too fast, the information rate can be high, but the refractory period stops front propagation and when the pulses are too slow, the information rate is low but the propagation is faithful.

Overall, this is an interesting study. However, the authors describe semi-qualitative phenomena without much experimental corroboration (including analysis of past experiments). Given that this manuscript does not identify nondimensional parameters important for the phenomena observed (propagation failure and information transfer) and that it does not explain/fit to previous experimental data, I would like the authors to expand their discussion further to make their manuscript more relevant for the audience. I have 2-3 suggestions that in my opinion will improve the manuscript significantly.

1. Information transfer calculation: It is perhaps better to have the inter-pulse-interval as a random variable rather than a fixed periodic value. This way, the authors can compute the mutual information rate between the input Markov process and the output Markov process. Perhaps, the authors can discuss this.

2. Topology of the network: The topology of the network is quite restrictive. For example, if I understand correctly, once excited, a cell must undergo the full transition to get back to the Q state. That is, the E state is not allowed to go back to the Q state or the Q state is not allowed to skip the E and the I states and directly jump to the R state. The authors should discuss how a less restrictive topology will affect their results.

3. Heterogeneity: My biggest concern with the work is the lack of heterogeneity. It has been repeatedly shown that cell to cell variability in RTK signaling pathways is dominated by parameter heterogeneity and not by stochasticity (e.g. Gross et al. Cell Systems 2019, Goetz et al. eLife 2024). Yet, the authors assume identical rate parameters for all cells. I suspect that heterogeneity can either hinder or enhance information transfer. I also understand that these calculations are not in the scope of the current work. Therefore, I would like the authors to at least discuss the role of heterogeneity.

Reviewer #3: See attached.

**Have the authors made all data and (if applicable) computational code underlying the findings in their manuscript fully available?**

Reviewer #1: Yes

Reviewer #2: Yes

Reviewer #3: **No: ** If accepted, authors should make simulation and analysis codes available.

PLOS authors have the option to publish the peer review history of their article (what does this mean? ). If published, this will include your full peer review and any attached files.

**Do you want your identity to be public for this peer review?** For information about this choice, including consent withdrawal, please see our Privacy Policy .

Reviewer #1: **Yes: ** Tsuyoshi Hirashima

Reviewer #2: No

Reviewer #3: **Yes: ** Peter Thomas
---

## [Decision Letter · Decision Letter 1]

14 Jan 2025

PCOMPBIOL-D-24-01158R1

Information transmission in a cell monolayer: A numerical study

PLOS Computational Biology

Dear Dr. Lipniacki,

Thank you for submitting your manuscript to PLOS Computational Biology. After careful consideration, we feel that it has merit but does not fully meet PLOS Computational Biology's publication criteria as it currently stands. Therefore, we invite you to submit a revised version of the manuscript that addresses the points raised during the review process.

Please submit your revised manuscript within 30 days Mar 16 2025 11:59PM. If you will need more time than this to complete your revisions, please reply to this message or contact the journal office at ploscompbiol@plos.org. Please include the following items when submitting your revised manuscript:

We look forward to receiving your revised manuscript.

Kind regards,

Oleg A Igoshin

Academic Editor

PLOS Computational Biology

Stacey Finley

Section Editor

PLOS Computational Biology

**Additional Editor Comments:**

While most of the technical concenrns of the prior reviews were successfully addressed, there still remaining a question on how relevant the measures of information are for the biology of the ERK/MAPK system. I suggest making a paragraph on the discussion discussing this in the context of the literature. Use questions from the Reviewer 3 as a guidance to structure these section.

**Journal Requirements:**

1) Thank you for uploading your study's underlying data set. We notice that there is a  GPL-3.0 license on your data. We would encourage you to consider using a license that is no more restrictive than CC BY, in line with PLOS’ recommendation on licensing (http://journals.plos.org/plosone/s/licenses-and-copyright).

**Reviewers' comments:**

Reviewer's Responses to Questions

**Comments to the Authors:**

**Please note that one of the reviews is uploaded as an attachment.**

Reviewer #1: The revisions have substantially improved the clarity, reproducibility, and overall quality of the study. I would like to express my appreciation for your thorough responses to my comments and the detailed revisions made to the manuscript. Overall, the revisions have strengthened the manuscript considerably, addressing my primary concerns. I believe this version is now suitable for publication.

Reviewer #2: The authors have addressed all my questions and made satisfactory changes.

Reviewer #3: See attached report.

**Have the authors made all data and (if applicable) computational code underlying the findings in their manuscript fully available?**

Reviewer #1: Yes

Reviewer #2: Yes

Reviewer #3: None

PLOS authors have the option to publish the peer review history of their article (what does this mean? ). If published, this will include your full peer review and any attached files.

**Do you want your identity to be public for this peer review?** For information about this choice, including consent withdrawal, please see our Privacy Policy .

Reviewer #1: **Yes: ** Tsuyoshi Hirashima

Reviewer #2: No

Reviewer #3: **Yes: ** An anonymous reviewer

**Figure resubmission:**
---

## [Editor Report · Decision Letter 2]

3 Feb 2025

Dear Prof. Lipniacki,

We are pleased to inform you that your manuscript 'Information transmission in a cell monolayer: A numerical study' has been provisionally accepted for publication in PLOS Computational Biology.

Best regards,

Oleg A Igoshin

Academic Editor

PLOS Computational Biology

Stacey Finley

Section Editor

PLOS Computational Biology

---

## [Editor Report · Acceptance letter]

PCOMPBIOL-D-24-01158R2

Information transmission in a cell monolayer: A numerical study

Dear Dr Lipniacki,

I am pleased to inform you that your manuscript has been formally accepted for publication in PLOS Computational Biology. Your manuscript is now with our production department and you will be notified of the publication date in due course.

With kind regards,

Anita Estes
